# Acromial and Scapular Spine Fractures following Reverse Total Shoulder Arthroplasty—A Systematic Review of Fixation Constructs and Techniques

**DOI:** 10.3390/jcm11237025

**Published:** 2022-11-28

**Authors:** J. Tristan Cassidy, Alexander Paszicsnyek, Lukas Ernstbrunner, Eugene T. Ek

**Affiliations:** 1Melbourne Orthopaedic Group, 33 The Avenue, Windsor, Melbourne, VIC 3181, Australia; 2Department of Orthopaedic Surgery and Traumatology, General Hospital Oberndorf, Paracelsusstraße 37, 5110 Oberndorf, Austria; 3Department of Orthopaedic Surgery, Royal Melbourne Hospital, 300 Grattan Street, Parkville, Melbourne, VIC 3050, Australia; 4Department of Biomedical Engineering, University of Melbourne, Parkville, Melbourne, VIC 3010, Australia; 5Department of Surgery, Monash Medical Centre, Monash University, Melbourne, VIC 3181, Australia

**Keywords:** reverse total shoulder arthroplasty, acromion fracture, scapular spine fracture, fixation techniques, open reduction and internal fixation

## Abstract

Fractures of the acromion and the scapular spine are established complications of reverse shoulder arthroplasty (RSA), and when they occur, the continuous strain by the deltoid along the bony fragments makes healing difficult. Evidence on treatment specific outcomes is poor, making the definition of a gold standard fixation technique difficult. The purpose of this systematic review is to assess whether any particular fixation construct offers improved clinical and/or radiographic outcomes. A systematic review of the literature on fixation of acromial and scapular spine fractures following RSA was carried out based on the guidelines of PRISMA. The search was conducted on PubMed, Embase, OVID Medline, and CENTRAL databases with strict inclusion and exclusion criteria applied. Methodological quality assessment of each included study was done using the modified Coleman methodology score to asses MQOE. Selection of the studies, data extraction and methodological quality assessment was carried out by two of the authors independently. Only clinical studies reporting on fixation of the aforementioned fractures were considered. Fixation construct, fracture union and time to union, shoulder function and complications were investigated. Nine studies reported on fixation strategies for acromial and scapular spine fractures and were therefore included. The 18 reported results related to fractures in 17 patients; 1 was classified as a Levy Type I fracture, 10 as a Levy Type II fracture and the remaining 7 fractures were defined as Levy Type III. The most frequent fixation construct in type II scapular spine fractures was a single plate (used in 6 of the 10 cases), whereas dual platin was the most used fixation for Levy Type III fractures (5 out of 7). Radiographic union was reported in 15 out of 18 fractures, whereas 1 patient (6.7%) had a confirmed non-union of a Levy Type III scapular spine fracture, requiring revision fixation. There were 5 complications reported, with 2 patients undergoing removal of metal and 1 patient undergoing revision fixation. The Subjective Shoulder Value and Visual Analogue Scale pain score averaged 75% and 2.6 points, respectively. The absolute Constant Score and the ASES score averaged 48.2 and 78.3 points, respectively. With the available data, it is not possible to define a gold standard surgical fixation but it seems that even when fracture union can be achieved, functional outcomes are moderate and there is an increased complication rate. Future studies are required to establish a gold standard fixation technique.

## 1. Introduction

The inherent medialization of the center of rotation and lengthening of the arm in reverse total shoulder arthroplasty (RSA) leads to an increase in deltoid moment arm, but also in increased strain on the acromion and scapular spine along the deltoid origin [1]. Acromion and scapular spine fractures are established complications of RSA, and when they occur, the continuous strain by the deltoid along the bony fragments makes healing difficult [2,3,4].

Several RSA-specific factors influence acromial/scapular spine strain, with glenoid lateralization being one of the main factors leading to increased strain [5]. Thus the growing trend towards increased lateralization of the center of rotation (COR) may exacerbate the difficult problem of acromion or scapular spine fractures following RSA [2,6].

The reported incidence of acromial and/or scapular spine fractures after RSA has shown marked variation between studies and ranges between 0.8 and 11.2% [7,8,9,10]. Both the Australian and United Kingdom joint registries confirm rising numbers of RSAs [2,6]. The rising implantation numbers and widening of indications for RSA suggest that shoulder surgeons will treat increasing numbers of acromial and scapular spine fractures in RSA patients [11]. It is important to identify the most reliable fixation technique for treating these fractures.

Unfavorable design features such as glenoid-sided lateralization make acromial fractures challenging to treat [12,13]. Other structural factors which may undermine healing include base-plate screws potentially weakening the scapular spine, poor bone stock and plates being positioned on the tension side of the fracture (increasing screw cut-out risk) [14]. Thus fixation of acromial/scapular spine fractures following RSA is both complex and technically demanding [15]. The literature contains an assortment of proposed fixation constructs varying in both the mode of fixation, such as plate fixation vs. tension band, as well as the individual fixation construct, such as number of plates, and use of specified plates (e.g., locking plates) [7,16,17]. The evidence on fixation technique specific outcomes is poor, making the declaration of a gold standard treatment difficult. The purpose of this systematic review is to analyze the current literature regarding the fixation methods of acromial/scapular fractures following RSA and assess whether any particular construct offers improved clinical and/or functional outcomes.

## 2. Materials and Methods

### 2.1. Search Strategy

The Preferred Reporting Items of Systematic Reviews and Meta-Analysis (PRISMA) statement provided the guidance for this systematic review. A systematic search was conducted of the PubMed, Embase, OVID Medline and CENTRAL (Cochrane Central Register of Controlled trials) databases. The following keywords were used for the search: “reverse shoulder arthroplasty”, “reverse total shoulder prosthesis”, “reverse shoulder prosthesis”, “reverse total shoulder arthroplasty” were combined with “acromial fracture”, “acromial insufficiency fracture”, “acromial stress fracture”, as well as “scapular spine fracture”, “scapular spine insufficiency fracture”, “scapular spine stress fracture”.

### 2.2. Selection Process

Two authors (AP, JTC) independently screened titles, abstracts, and full-texts using the predefined inclusion and exclusion criteria. In cases of discrepancy, the senior author (LE) was consulted until a final consensus was reached. Studies reporting on fixation after acromial or scapular spine fracture following reverse total shoulder arthroplasty were chosen based on the following inclusion criteria:(1)clinical studies (clinical = in vivo)(2)studies reporting specifically on the treatment of acromial or scapular spine fractures(3)studies in German or English language(4)studies published on/after 1 January 1990

We excluded editorial comments, review articles, conference proceedings, studies not reporting on specific treatment after acromial/scapular spine fracture following RSA and studies that were not in English or German language.

### 2.3. Data Extraction/Interpretation

The relevant information regarding the study characteristics was collected by a single author (JTC) using a predetermined data sheet. The study characteristics included the study design, level of evidence, methodological quality of evidence (MQOE), population, union, clinical outcome measures, and follow-up time points. Extracted clinical outcomes analyzed included: (1) clinically and radiologically confirmed fracture union; (2) Subjective Shoulder Value (SSV) and Visual Analogue Scale (VAS) pain score; (3) absolute Constant Score and American Shoulder and Elbow Surgeons Shoulder Score (ASES); (4) range of motion data. Clinical fracture union was defined as a painless and functional shoulder without obvious clinical deformation of the acromion and/or scapular spine [18]. Acromial and scapular spine fractures were classified according to the Levy classification [19]. Type I involves a fracture through the midpart of the acromion caused by the anterior and middle deltoid origin. Type II was defined as a fracture caused by the entire middle deltoid segment and portion of posterior deltoid origin. Type III fractures involve the entire middle and posterior deltoid origin (Figure 1).

### 2.4. Study Quality Assessment

The Quality assessment was conducted using the guidelines created by the Oxford Centre for Evidence Based Medicine. This aided in the evaluation of the level of evidence, whereas the modified Coleman methodology score was used to evaluate the MQOE [20]. According to these previous guidelines, studies were considered excellent quality if they scored 85 to 100; good quality, 70 to 84; fair quality, 55 to 69; and poor quality, less than 55. Two reviewers (AP, JTC) assessed the studies independently and in case of inter-rater disagreement, one of the senior authors (LE) was consulted until final consensus was reached.

## 3. Results

### 3.1. Search Results

This study was registered with the PROSPERO (CRD42022360274). The initial literature search resulted in a total of 885 studies. After removal of duplicates, 128 articles were screened for inclusion and exclusion criteria, and 17 unique studies were evaluated and full texts were assessed for eligibility.

Three studies were excluded as the specific treatment for acromial/scapular spine fracture was not mentioned [21,22,23]. Three studies did not report on acromial/scapular spine fracture in the context of RSA, one study was not in German or English language, and one study did not report on specific outcomes after treatment of acromial/scapular spine fracture [24,25,26,27,28]. One study was identified from review references of the identified articles [29]. In total, nine clinical studies with 18 acromion or scapular spine fractures met the inclusion and exclusion criteria and were included in this review [16,17,18,29,30,31,32,33,34]. A detailed flow diagram of the search results is provided under Figure 2.

### 3.2. Study Quality Assessment

In this review, all studies were ranked as poor quality according to the MQOE. The Maximum Score was 48 and the minimum 10 with a mean score of 29.8 points. These low scores are caused by a limited number of patients as most publications are level IV evidence case reports or retrospective case series. A detailed list of the characteristics and Quality Assessment scores of the MQOE is found in Table 1.

### 3.3. Study Characteristics and Patient Demographic Characteristics

Nine clinical studies reported a total of 18 acromion/scapular spine fractures in 17 patients (Table 1). Thus, the mean number of patients per study was two (range, 1–5 patients). The majority of patients were female (83.3), the average age was 74.2 years (range, 63–83 years) and the mean follow-up was 13.4 months (range, 2–24 months). Time from RSA to fracture fixation was not reported in two case reports [16,29]. These cases underwent fixation with a lateral locking plate with two lag screws and a 1/3 tubular plate, respectively (Table 2). For the remaining seven studies, the mean time from index RSA to fracture fixation was 31.8 months (range, 4–115 months). Study characteristics and patient demographic characteristics are shown in Table 1.

**Table 1 jcm-11-07025-t001:** Study Characteristics and Patient Demographic Characteristics.

Authors	n = Fracture (n = Patient)	Prospective or Retrospective	LOE	MQOE Score	Female Patients	Age, yr	Follow-Up, mo
Bauer et al., 2020 [32]	1 (1)	Retrospective	IV	38	1	74	12
Cho et al., 2019 [16]	1 (1)	Retrospective	IV	48	0	63	18
Debeer et al., 2005 [29]	1 (1)	Retrospective	IV	28	1	83	2
Hess et al., 2018 [17]	3 (3)	Retrospective	IV	34	3	74.7	12
Khwaja et al., 2020 [34]	1 (1)	Retrospective	IV	22	1	72	12
Kim et al., 2021 [30]	2 (1)	Retrospective	IV	26	0	79	24
Rouleau et al., 2013 [33]	1 (1)	Retrospective	IV	22	1	71	18
Toft et al., 2019 [18]	5 (5)	Retrospective	IV	30	5	75.8	13.4
Wahlquist et al., 2011 [31]	3 (3)	Retrospective	IV	10	2	72	13

LOE: level of evidence; MQOE: methodological quality of evidence.

### 3.4. Fracture Characteristics and Fixation Constructs

Within the identified cohort of 18 fractures, one was classified as a Levy Type I fracture [16], 10 were Levy Type II fractures [17,30,31,32,33], and the remaining 7 fractures were defined as Levy Type III [18,29,34]. Fractures were classified according to Levy et al. in two studies (four fractures) [13,17,19,34], with Levy classification ascertained by the authors for the other seven studies.

Five out of seven Levy Type III fractures underwent dual plate fixation. The remaining two Levy type III fractures received a single plate fixation construct [29,34]. Three out of ten Type II fractures underwent dual plate fixation, six Type II fractures underwent single plating and the remaining patient with a Type II fracture received a tension band wire construct. The single patient with a Type I fracture underwent single plating (Table 2).

When stratifying the data according to fixation construct, the following results were pooled (Table 3): Eight patients were treated with a dual plate configuration [18,31,32,33]. In five patients a superiorly placed locking plate in combination with a caudally placed 1/4 tubular plate was used. The locking plates used in these patients were a 3.5 mm locking compression plate (LCP) in 1 patient, a variable angle olecranon plate in another patient a variable angle lateral distal humerus plate in 3 patients (all plates DePuy Synthes, Oberdorf, Switzerland) [18]. Another fracture was treated using a 2.4 mm LCP plate caudally in combination with a lateral clavicle locking plate (DePuy Synthes, Oberdorf, Switzerland) placed posteriorly [32] superiorly on the scapular spine and acromion. One patient received treatment using a lateral clavicle locking plate in combination with a reconstruction plate (both DePuy Synthes, Oberdorf, Switzerland) in a 90/90 (posterior/caudal) configuration [33]. For the remaining fracture, a combination of a 3.5 mm LCP plate superiorly and a 2.7 mm LCP plate posteriorly (both Zimmer, Warsaw, IN, USA) was used [31].

Four patients underwent fixation using a single lateral clavicle locking plate [16,30,34] (Acumed, Hillsboro, OR, USA). Three patients were treated with a cruciform pilon locking plate (Depuy Synthes, Oberdorf, Switzerland) [17]. In one case, the scapular spine fracture was fixed using a 7-hole 1/3 tubular plate [29]. Another scapular spine fracture was fixed using a reconstruction plate [31]. For the last construct, a combination of tension band wire and a lag screw was used [31].

Bone graft (substitute) was used in a total of five cases [18,31]. In four cases of Type III fractures, cancellous bone graft from the iliac crest was used and was augmented with bicortical iliac crest bone graft in two out of these four cases [18]. One Levy Type II fracture patient received demineralized bone matrix augmentation [31]. Digital abduction force testing was recorded in one case as part of the Constant score and recorded at 16 pounds without pain [32], despite the occurrence of a plate-adjacent fracture (Levy type I) which had also united at one year.

### 3.5. Fracture Union and Reported Outcomes

Fracture healing was confirmed clinically in 17 out of 18 fractures (94.4%) in nine studies [16,17,18,29,30,31,32,33,34]. Fifteen fractures had plain radiographs to assess union [16,17,18,30,31,32,33,34]. Fourteen out of fifteen fractures had a confirmed union, with one reported non-union (6.7%) of a Levy Type III scapular spine fracture, resulting in a downward sloped acromion requiring revision fixation [31]. One was a Levy Type I fracture, eight of these cases were Type II fractures and six were a Levy Type III fractures.

Time to union was reported for five patients in three studies [30,31,32]. The mean time to union for the reported five patients was 8.5 months (range 2–24 months). Detailed information about fracture location and union rates is provided in Table 3.

The SSV score was reported in 8 patients (4 studies [16,18,30,32]) and the VAS pain score was reported for 8 fractures (3 studies [16,18,30]) with an average of 75% and 2.6 points, respectively. The absolute Constant Score was reported in 2 studies with a total of 6 patients [18,32]; the ASES score was reported in 2 patients (2 studies [16,30]) and averaged 48.2 points and 78.3 points, respectively. Range of motion was reported in 13 patients (6 studies [16,18,29,30,31,32]) and averaged for forward elevation at 113°, for abduction at 105° and for external rotation at 33°.

### 3.6. Complications

There were five complications (27%) reported [17,18,31,32]. In one patient, loss of fixation was reported that needed revision surgery [31]. The initial fracture was classified as a Levy Type II which was fixed with a reconstruction plate that failed eventually. No details about revision surgery are provided but union at final follow-up was achieved. Another patient had a periprosthetic fracture 3 weeks after fixation of a Levy type II fracture [32]. The periprosthetic fracture was at the lateral aspect of the acromion (Levy Type I) just lateral to the plates. The new fracture healed without further surgery but resulted in a caudally tilted acromion and a prominent plate so that the patient had to undergo removal of the caudally positioned 2.4 LCP plate once union was confirmed. The clinical outcome on the Constant score was good (not age corrected; 67 points; good strength) and the second plate (lateral clavicle locking plate) was left in-situ due to concerns of stability. A second patient underwent removal of metal, although the exact circumstances were not further specified regarding why the posteriorly positioned cruciform pilon plate had to be removed [17].

The other two complications included one screw loosening and migration and one iatrogenic pneumothorax [18], of which none needed revision surgery. Detailed information about complications is presented in Table 3.

**Table 2 jcm-11-07025-t002:** Fixation constructs according to fracture classification.

Authors	Patient- Number	Primary Indication for RSA	Result of Trauma (Y/N)	Time between RSA & ORIF, Months	Fixation Construct	Location of Plate(s)	Graft (Y/N)
**Levy Type I [19]**							
Cho et al., 2019 [16]	1	Failed massive cuff repair	Y	4	2 lag screws (4.5 mm cannulated) 1 lateral clavicle locking plate	Posterior scapular spine	N
**Levy Type II [19]**							
Hess et al., 2018 [17]	1	Massive cuff tear	N	6	Cruciform pilon locking plate & lag screw	Posterior scapular spine	N
	2	Massive cuff tear	Y	23	Cruciform pilon locking plate & lag screw	Posterior scapular spine	N
	3	Massive cuff tear	N	8	Cruciform pilon locking plate & lag screw	Posterior scapular spine	N
Bauer et al., 2020 [32]	1	Primary glenohumeral osteoarthritis	Y	1.25	1 Lateral clavicle locking plate 1 2.4 mm LCP plate	Posterior scapular spine (clavicle plate), caudal scapular spine (LCP plate)	N
Kim et al., 2021 [30]	1	Failed massive cuff repair	N	Not reported	Lateral clavicle locking plate	Posterior scapular spine	N
	1	Failed massive cuff repair	N	Not reported	Lateral clavicle locking plate	Posterior scapular spine	N
Rouleau et al., 2013 [33]	1	Rotator Cuff Arthropathy	Y	6	1 Lateral clavicle locking plate 1 Reconstruction plate	Superior scapular spine Posterior scapular spine	N
Wahlquist et al., 2011 [31]	1	Failed cuff repair	N	11	1 3.5 mm LCP plate 1 2.7 mm LCP plate	Superior scapular spine Posterior scapular spine	N
	2	Massive cuff tear	N	12	1 Lag Screw 1 Tension band	Details not provided	Y
	3	Failed cuff repair	N	24	1 Reconstruction plate	Details not provided	N/A
**Levy Type III [19]**							
Khwaja et al., 2020 [34]	1	Rotator cuff arthropathy	N	60.5	Lateral clavicle locking plate	Posterior scapular spine	N
Debeer et al., 2005 [29]	1	Rotator Cuff Arthropathy	Y	9	1 1/3 Tubular plate	Posterior Scapular Spine	N
Toft et al., 2019 [18]	1	Massive cuff tear with avascular necrosis	N	4.5	1 3.5 mm LCP plate 1 ¼ Tubular plate	Posterior scapular spine Caudal scapular spine	Y
	2	Failed fracture fixation	Y	97.3	1 VA Olecranon plate 1 ¼ Tubular plate	Posterior scapular spine Caudal scapular spine	N
	3	Rotator Cuff Arthropathy	Y	36	1 VA Lateral distal humerus plate 1 ¼ Tubular plate	Posterior scapular spine Caudal scapular spine	Y
	4	Dislocation with massive cuff tear	Y	68	1 VA Lateral distal humerus plate 1 ¼ Tubular plate	Posterior scapular spine Caudal scapular spine	Y
	5	Rotator cuff arthropathy	N	115	1 VA Lateral distal humerus plate 1 ¼ Tubular plate	Posterior scapular spine Caudal scapular spine	Y

RSA, Reverse Shoulder Arthroplasty; Levy, Levy Classification [19]; ORIF, Open reduction internal fixation; LCP, locking compression plate; VA, variable angle.

**Table 3 jcm-11-07025-t003:** Fracture location, union and complications stratified by fixation construct.

Authors		Additional Fixation	Levy Zone [19]	Mechanism of Injury	Location of Plate	Union	Complication
**Single Plate**							
	Lateral clavicle locking plate						
Cho et al., 2019 [16]		2× lag screws	I	Traumatic	Posterior Scapular Spine	Y	Nil reported
Kim et al., 2021 [30]		Nil	II	Atraumatic	Posterior Scapular Spine	Y	Nil reported
		Nil	II	Atraumatic	Posterior Scapular Spine	Y	Nil reported
Khwaja et al., 2020 [34]		Nil	III	Traumatic	Posterior Scapular Spine	Y	Nil reported
	Cruciform pilon locking plate						
Hess et al., 2018 [17]		2× lag screws	II	Atraumatic	Posterior Scapular Spine	Y	Nil reported
		2× lag screws	II	Traumatic	Posterior Scapular Spine	Y	Nil reported
		2× lag screws	II	Atraumatic	Posterior Scapular Spine	Y	Plate removal
	Reconstruction plate						
Wahlquist et al., 2011 [31]		Nil	II	Atraumatic	Not stated	N	Loss of fixation requiring revision
	1/3 Tubular plate						
Debeer et al., 2005 [29]		Nil	III	Traumatic	Posterior Scapular Spine	Y	Nil reported
**Dual Plates**							
	Lateral clavicle locking plate						
Bauer et al., 2020 [32]		2.4 mm LCP plate	II	Traumatic	Posterior scapular spine (Clavicle plate), Caudal scapular spine (LCP plate)	Y	Periprosthetic fracture Plate removal
Rouleau et al., 2013 [33]		Reconstruction plate	II	Traumatic	Superior scapular spine (Clavicle plate), Posterior scapular spine (Reconstruction plate)	Y	Nil reported
	¼ Tubular Plate						
Toft et al., 2019 [18]		3.5 mm LCP plate	III	Atraumatic	Superior scapular spine (1/4 Tubular plate), Posterior scapular spine (LCP plate)	Y	Screw loosening and migration
		VA Olecranon plate	III	Traumatic	Superior scapular spine (1/4 Tubular plate), Posterior scapular spine (Olecranon plate)	Y	Nil reported
		VA Lateral distal humerus plate	III	Traumatic	Superior scapular spine (1/4 Tubular plate), Posterior scapular spine (Distal humerus plate)	Y	Iatrogenic pneumothorax
		VA Lateral distal humerus plate	III	Traumatic	Superior scapular spine (1/4 Tubular plate), Posterior scapular spine (Distal humerus plate)	Y	Nil reported
		VA Lateral distal humerus plate	III	Atraumatic	Superior scapular spine (1/4 Tubular plate), Posterior scapular spine (Distal humerus plate)	Y	Nil reported
	3.5 mm LCP plate						
Wahlquist et al., 2011 [31]		2.7 mm LCP plate	II	Atraumatic	Superior scapular spine (Non-Locking Plate 3.5 mm (7-hole), Posterior scapular spine (Non-locking 2.7 mm (8 hole))	Y	Nil reported
**Alternative**							
	Tension Band Wire						
Wahlquist et al., 2011 [31]		Lag screw	II	Atraumatic	N/A	Y	Nil reported

LCP, locking compression plate; VA, variable angle.

## 4. Discussion

There is a general paucity of knowledge about treatment of acromial and scapular spine fractures following RSA. The included studies showed a high degree of heterogeneity and the included sample size is underpowered. It is therefore difficult to draw a meaningful conclusion about which fixation construct may reveal the most reliable outcome in this difficult clinical scenario. Although we were unable to determine the fixation construct with the best clinical performance, our study of pooled results suggests that using a single locking plating for lateral acromion and acromial base fractures (Levy Type I and II), and a dual plating construct for more medial fractures (Levy Type II and III) delivers a high union rate and a reasonably low complication rate. Together with the current biomechanical knowledge about acromion and scapular spine fractures following RSA [5] and plating techniques [35], it seems that dual plating reveals a stronger fixation construct than single plating and may yield superior outcomes in more medial fractures, although the latter is yet to be proven.

Notwithstanding, there is an ongoing debate whether single or dual plating should be preferred [35]. Of the included studies, eight fractures were fixed with a dual plating construct [18,31,32,33] and in nine fractures a single plate was used [16,17,29,30,31,34] (the remaining patient had a tension band wire construct [31]). Apart from two patients [29,34], dual plating was the preferred fixation method for Levy Type III fractures [18,32], whereas single plating was the most common technique for more lateral fractures [16,17,30,31,34]. More medial fractures (i.e., Levy Type III) are harder to fix for reasons such as poor bone stock and the broad deltoid origin which is a constant displacing force. In a study by Katthagen et al. [25], dual plating with two LCP plates revealed superior fixation strength compared with a single LCP plate. This was confirmed by a recent study by Hollensteiner et al. [35], who showed that for Levy Type III fractures, dual plating using two LCP plates in a orthogonal configuration is stronger than a single plate positioned caudally or posteriorly. When only one plate is used for fracture fixation, a LCP plate has shown to have superior fixation strength compared to a single reconstruction or lateral locking plate, for Levy Type III fractures [26].

Positioning of the plates has also been investigated and it has been shown that plates can be placed superiorly (along the supraspinatus fossa), posteriorly (along the scapular spine) or caudally (along the infraspinatus fossa) [13,25,26,35]. For single plating, applying the plate over the posterior aspect of the scapular spine might be the strongest option [13]. When dual plating is performed, placing the plates over the posterior and caudal aspect of the scapular spine might provide the strongest fixation [25,35].

The aforementioned biomechanical aspects, union rates, outcome scores and complication rates are yet to be further studied in order to define a gold standard fixation technique. In our systematic review, radiographically confirmed union was reported in 14 out of 15 patients assessed, and except for one patient with a non-union [31], all of the remaining 14 patients had clinically confirmed union. It is obvious that there is a lack of radiographic documentation of fracture healing in the literature but of those considered as clinically healed, pain levels were acceptable with a pooled VAS pain score of 2.6 [16,18,30]. On the other hand, pooled SSV, absolute Constant Score and ASES score were moderate [16,18,30,32], and below the scores reported for patients following RSA without acromion or scapular spine fractures [36,37,38,39,40,41,42]. The pooled range of motion values were also slightly below the reported results in the literature, which leads to the assumption that even if acromion or scapular spine fractures are reported to be healed (clinically), one might anticipate that functional outcome will be compromised.

The complication rate in our systematic review was 27% and included one loss of fixation of a Levy Type III fracture which was fixed initially with a single reconstruction plate that ultimately healed after revision fixation [31]. Another two patients underwent plate removal, with at least one case for irritating metalware [32] and the second not further specified [17]. One must therefore also anticipate a higher complication rate in patients receiving RSA for acromial or scapular spine fractures that underwent surgical fixation.

## 5. Limitations

The most important limitation to this review is the low numbers of patients identified within the literature obviating the possibility of any firm data-driven conclusions. Outcomes reported were incomplete (radiographic union) and heterogenous (functional outcome scores). With a mean MQOE of 29.8, the quality of the available clinical studies was poor as classified by Coleman et al. [20]. Furthermore, there are several more general limitations inherent to systematic reviews, including publication bias, search bias, selection bias, and heterogeneity of results. Selection bias may have been introduced due to the fact that all of the included studies were retrospective and none of the studies were controlled. Surgical techniques were varied both within and between included studies, which could potentially have influenced outcomes. PROSPERO registration was conducted after the completion of this systematic review. However, the protocol of this study was followed strictly and no changes have been made to it, neither during nor before submission to PROSPERO for registration.

## 6. Conclusions

Literature about surgical fixation of acromial and scapular spine fractures following RSA is limited. The included studies have a high degree of heterogeneity and are of poor quality, while the included sample size is underpowered. With the available data, it is not possible to define a gold standard surgical fixation but it seems that even when fracture union can be achieved, functional outcome is moderate and there is an increased complication rate. Future studies are required to establish a gold standard fixation technique.

## Figures and Tables

**Figure 1 jcm-11-07025-f001:**
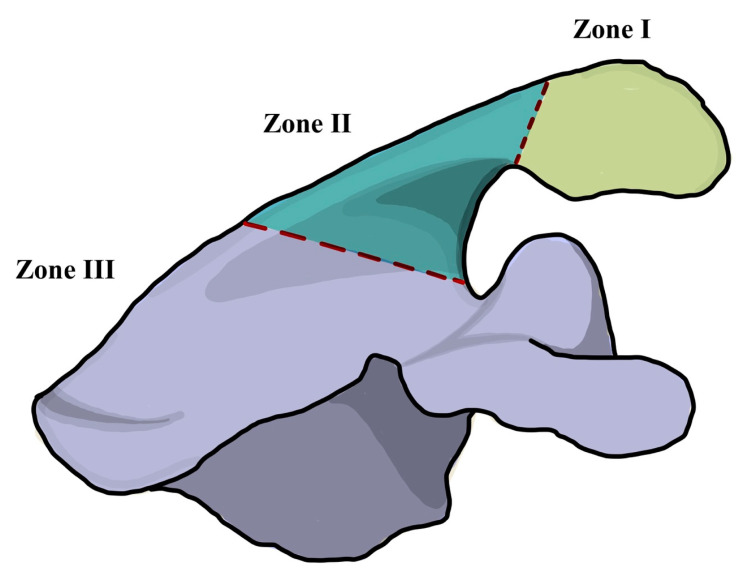
Illustration of the Levy classification for acromial stress fractures after reverse total shoulder arthroplasty [19]. Zone I involves a fracture through the midpart of the acromion caused by the anterior and middle deltoid origin. Zone II was defined as a fracture caused by the entire middle deltoid segment and portion of posterior deltoid origin. Zone III fractures involve the entire middle and posterior deltoid origin.

**Figure 2 jcm-11-07025-f002:**
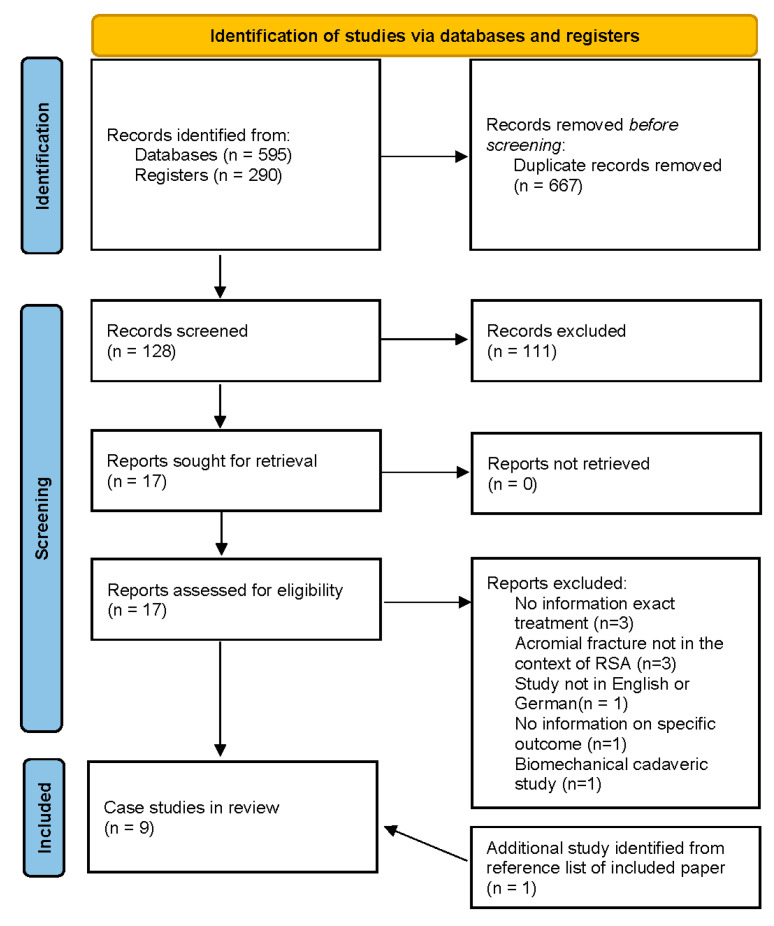
PRISMA flow diagram (updated version 2020) of the systematic search. RSA, reverse total shoulder arthroplasty.

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
