# Peer review of "Acromial and Scapular Spine Fractures following Reverse Total Shoulder Arthroplasty—A Systematic Review of Fixation Constructs and Techniques"

_jcm, 2022, doi:10.3390/jcm11237025_

Round 1

Reviewer 1 Report

Dear authors,

you try to perform a meta-analysis on a difficult and rarely reported topic. Although there are many reports in acromial and scapular spine fractures after RSA, most of them deal with conservative treatment and no high numbers are found for surgical repair. This is a known flaw of your meta-analysis that cannot be changed.

Reviewer 2 Report

Abstract:

Too wordy.  Shorten. 

Introduction.

Several statement regarding prosthetic design unrelated to this study.  86-90

The statement “unfavorable design features such as glenoid sided lateralization…”  has nothing to do with fixation constructs in these fractures and is opinion even though its referenced.  Recommend limiting introduction comments to the topic of the study.

Materials and Methods:

194-209 should be in methods section.

Results:

274-276-stated in methods

Include plate constructs in Table 2 and manufacturer information-reference table in body of report.

322-26.  Was union time 8.5 months for all cases or just the 5.  Confusing

350. I think the author meant a third patient-not second

316  Author states 17/18 healed but 2 had revision for fracture-clarify please

Discussion:

372-375  “High union rate”  while this is true that the analysis showed a high union rate the experience of many surgeons is otherwise-please clarify and be less definitive in this statement.  This is a difficult problem and has not been solved by this systematic review.  Complication rate is high-not low.

405  “radiographically confirmed union was inconsistently 405 reported in 14 out of 15 patients assessed”  this contradicts the prior statements on high union rate.

Reviewer 3 Report

I commend the authors to this well performed systematic review. Unfortunately, to date there is a reduced body of literature about successful fixation of these scapular spine fractures and a paucity of comparative studies of decent quality.

Comments:

57-59 Make this clearer:

The most frequent fixation construct in type II scapular spine fractures was a single plate in 6 of 10 cases, whereas……

78-80 The reference of the first statement should be changed:

The quoted paper by Zmistowski et al concludes that increased preop COR medialization was a risk factor for fracture.

Your statement is rather supported by the paper

Acromial Fractures Following Reverse Total Shoulder Arthroplasty: A Cohort Controlled Analysis” Schenk et al. Orthopaedics 2020

As described below several authors found that postop lateralization of the COR compared to the preop state may be more important than lengthening and an increase of the moment arm (-> glenoid sided lateralization as per Frankle and Levy/BIO-RSA as per Boileau). Glenoid lateralization reduces the moment arm. This is supported by the papers “Acromial Fractures Following Reverse Total Shoulder Arthroplasty: A Cohort Controlled Analysis by Kerrigan …..Athwal Shoulder and Elbow 2021 and Implant positioning in RSA has an impact on acromial stress Wong…Athwal, JSES 2016…..you develop all this correctly further down in the introduction

240-241

Box: call it: Case studies included in review (n=9)

251-253 this sentence is a bit awkward:

“These scores resulted of the limited number of specimen included resulting from most of the papers being case reports or retrospective case series”

Simplify it:

eg. These low scores are caused by a limited number of patients as most publications are level IV evidence case reports or retrospective case series. (delete the following sentence: all the included…)

258 ….included nine clinical case studies

293-295 Another fracture was treated using a 2.4mm LCP plate caudally in combination with a lateral clavicle locking plate (DePuy Synthes, Oberdorf, Switzerland) placed posteriorly32 superiorly on the spine and acromion.

335 Digital abduction force testing was recorded in one case as part of the Constant score and recorded at 16pounds without pain 32 (Bauer et al.) despite the occurrence of a plate-adjacent fracture (Levy type I) which had also united at one year.

343 typo: perirposthetic periprosthetic

344

The periprosthetic fracture was at the lateral aspect of the acromion just lateral to the plates (add: Levy type I).

348-349

(add) The clinical outcome on the Constant score was good (not age corrected; 67points; good strength) and the second plate (lateral clavicle locking plate) was left in-situ due to concerns of stability.

Table 2:

Bauer et al.

Posterior  Superior scapular spine (clavicle plate), caudal scapular spine (LCP plate)

Bauer et al.

“The SS fracture was manually reduced and the plate was applied to the superior aspect of the SS with a medial cortical screw and subsequent lateral acromial and medial locking screws. To create a 90°-double plating construct as previously described by Rouleau and Gaudelli,1 a 2.4 mm straight LCP (Synthes/compact foot set) was bent to the shape of the SS and first medially compressed with a cortical screw before being fixed with locking screws on either side of the fracture.”

Table 3:

Bauer et al.

Posterior  Superior scapular spine (clavicle plate)
